# Dietary Polysaccharides as Modulators of the Gut Microbiota Ecosystem: An Update on Their Impact on Health

**DOI:** 10.3390/nu14194116

**Published:** 2022-10-03

**Authors:** Ana I. Álvarez-Mercado, Julio Plaza-Diaz

**Affiliations:** 1Instituto de Investigación Biosanitaria IBS.GRANADA, Complejo Hospitalario Universitario de Granada, 18014 Granada, Spain; 2Department of Biochemistry and Molecular Biology II, School of Pharmacy, University of Granada, 18071 Granada, Spain; 3Institute of Nutrition and Food Technology, Biomedical Research Center, University of Granada, 18016 Armilla, Spain; 4Children’s Hospital of Eastern Ontario Research Institute, Ottawa, ON K1H 8L1, Canada

**Keywords:** dietary polysaccharides, gut microbiota, diet, health, fiber

## Abstract

A polysaccharide is a macromolecule composed of more than ten monosaccharides with a wide distribution and high structural diversity and complexity in nature. Certain polysaccharides are immunomodulators and play key roles in the regulation of immune responses during the progression of some diseases. In addition to stimulating the growth of certain intestinal bacteria, polysaccharides may also promote health benefits by modulating the gut microbiota. In the last years, studies about the triad gut microbiota–polysaccharides–health have increased exponentially. In consequence, in the present review, we aim to summarize recent knowledge about the function of dietary polysaccharides on gut microbiota composition and how these effects affect host health.

## 1. Introduction

### 1.1. Food Polysaccharides, an Overview

Carbohydrates are divided in several categories based on their number of sugar units: (a) monosaccharides have one sugar molecule; (b) disaccharides have two sugar molecules; (c) oligosaccharides have three to ten sugar units and may be produced by the breaking down polysaccharides; and (d) polysaccharides are macromolecules of monosaccharides consisting of more than ten units [1]. Polysaccharides are the major components of dietary fiber [2]. They bind to bile acids in the small intestine, thereby lowering serum cholesterol and normalizing blood lipid levels [3]. Most of the structures of polysaccharides are associated with numerous biological benefits for gut health and are frequently found in more complex structures that also contain digestible carbohydrates and proteins [4].

Food products contain polysaccharides derived from many sources, including farms, forests, oceans, fermentation vats, and chemical modification of natural polysaccharides, such as cellulose and starch [5]. Of the source and polysaccharide types, examples include algal (seaweed extracts) derived from agar, algins, carrageenans, and furcellaran, higher insoluble plants derived from cellulose, fruit extracts derived from pectin, corn starches, rice starches, wheat starches, beta-glucans, guar gum, locust bean gum, tara gum, psyllium seed gum, and tamarind seed polysaccharides [6].

For instance, hydrocolloids (plant-derived ingredients such as pectin, guar gum, locust bean gum, and konjac mannan) are a class of food ingredients mainly composed of polysaccharides and some proteins that are widely used in several food products [7]. Other polysaccharides are also commonly found in dietary products including starch, cellulose, chitosan, xyloglucan, glucan, xanthan, arabinoxylan, carrageenan, inulin, agar, and plant gums [8]. Indeed, starch is the second most abundant natural polysaccharide after cellulose and is the world’s primary source of food carbohydrates [9], while non-starch polysaccharides (NSPs) are non-glucan polysaccharides [10]. There are several hundred thousand monosaccharides units in NSPs that are linked through glycosidic bonds, making them more complex than starch [11]. The diverse categories of NSPs differ in terms of water solubility, size, and structure.

The ability of dissolved polysaccharides to thicken solutions and form gels is one of their most critical functional characteristics both in terms of formulation functionality and health-related functionality [12]. A large hydrodynamic volume of polysaccharides results in increased viscosity at low concentrations, while a small hydrodynamic volume results in decreased viscosity. As a consequence, high solubility (i.e., favorable interaction with the solvent which results in the polysaccharide expanding and a higher hydrodynamic volume) is beneficial for thickening [13]. Additionally, associative interactions may enhance the thickening properties of some modified starches and celluloses. A polysaccharide’s ability to form gels is dependent on its solubility. This is essential to the gel structure’s ability to hold water and the formation of a continuous network in the solution. It is therefore necessary for the polysaccharides to interact in some way in order to form associations [14]. In order for molecules to associate, they must exhibit either a hydrophobic effect, partial local crystallization, calcium bridges, or double or triple helices [2]. For instance, cellulose, galactomannans, xylans, xyloglucans, and lignin are water-insoluble fibers, while pectins, arabinogalactans, arabinoxylans, and -(1,3)(1,4)-D-glucans (-glucans) are water-soluble fibers [15].

As mentioned, polysaccharides are widely used in food technology and recognized for their bioactivity, which has been linked to a reduced risk of non-communicable chronic diseases [16]. NSPs and resistant starch (RS) are beneficial mediators of anti-inflammation, gut epithelial barrier protection, and immune modulation [17]. They possess antibacterial and anticancer properties. Polysaccharides also have antithrombotic, antioxidant, antiangiogenic, and antiviral properties [8]. The soluble part of dietary polysaccharides is related to an increase in transit time over the intestine. The insoluble fiber from dietary polysaccharides is linked to a decrease in transit time over the gastrointestinal tract. This is related to an augmentation in the excretion of bile acid and fecal bulk [18]. On the other hand, starch has a critical function in digestive processes because of the intermediation of ion exchange and holding water [19]. Additionally, bacterial polysaccharides, usually found in the cell wall, act as immune modulators. By interacting with gut microbes, host-derived polysaccharides protect host cells from pathogenic microbial neighbors, as well as affect overall intestinal health [17]. Indigestible but fermentable polysaccharides (termed prebiotics) can stimulate the growth and activity of beneficial bacteria in the colon. Accordingly, the inclusion of polysaccharides in the diet is therefore beneficial to the host metabolism, fat accumulation, and insulin resistance, among other benefits [8].

### 1.2. Food Polysaccharides and Gut Microbiota

The intestinal or gut microbiota is “the set of microbes that colonize our digestive tract and interact with each other and with the host” [20,21]. Indeed, the microbes that reside in our gut have a remarkable potential to influence physiology, both in disease and the health of the host. The gut microbiota modulates, directly or indirectly, most of our physiologic functions, including metabolic and pathogenic functions, as well as the immune system maturation [22]. The microbiome also encompasses all of the genetic information contained in the microbiota [23], creating a dynamic, interactive microecosystem capable of changing in time and scale, along with being integrated into macro-ecosystems including eukaryotic hosts, and being crucial to their health and functioning [24]. A gut ecosystem with a wide variety of species may be more resilient to environmental influences than one that lacks diversity, since functionally linked microbes within an intact ecosystem may be able to balance the function of other species that have become extinct. In consequence, a higher diversity is commonly regarded as an indicator of a healthy digestive system [25,26]. Thus, an equilibrated microbiota community frequently exhibits high taxonomic diversity, stable core microbiota, and high microbial gene richness [27,28]. In healthy conditions, the intestinal microbiota is stable, resilient, and interacts symbiotically with the host [27,28]. By contrast, an imbalance in gut microbiota composition and function (dysbiosis) has been linked to cardiovascular disease [29], cancer [30,31], respiratory diseases [32,33], diabetes [34], inflammatory bowel disease [35], brain disorders [36], chronic kidney disease [37], and liver disease [38], among others.

Physiological properties of the gastrointestinal tract are revealed by the composition of the microbiota in a given region, which is stratiform both transversely and longitudinally. Microbiota density and composition are influenced by nutritional, chemical, and immunological gradients along the gut [39]. The large intestine has high levels of oxygen, acids, and antimicrobials, as well as a longer transit time than the small intestine [40]. However, facultative anaerobes with the ability to adhere to epithelial or mucus surfaces are thought to survive in the large intestine, as are rapidly growing bacteria [40]. Besides, according to animal studies, the microbial community of the small intestine is essentially dominated by *Lactobacillaceae* (traditionally classified as oxygen-tolerant anaerobes) [41]. A diverse and dense bacteria community occurs in the colon, primarily anaerobes with the ability to utilize complex carbohydrates, which are undigested in the small intestine. It has been reported that *Prevotelaceae*, *Lachnospiraceae,* and *Rikenellaceae* constitute the majority of species in the colon [39,42].

There is a spatial preservation of microbiota diversity and composition in the colorectal mucosa region [43,44]. On the contrary, the compositions of the mucosal and fecal/luminal regions are drastically different [45]. *Bacteroidetes* are more abundant in fecal/luminal samples than in the mucosa samples. *Firmicutes,* specifically *Clostridium* cluster XIVa, are enriched in the mucus compared with the luminal/fecal regions [46,47].

Gut microbiota is also involved in the metabolism of choline, phosphatidylcholine, and carnitine and can produce trimethylamine-N-oxide (TMAO). Smooth muscle cells and endothelial cells can respond to TMAO by triggering the nuclear factor kappa-B (NF-κB) and mitogen-activated protein kinase (MAPK) signaling pathways [48]. Several recent reviews have faced this topic in detail elsewhere [49,50,51,52,53,54].

Concerning the interaction of gut microbiota–food polysaccharides, several dietary polysaccharides are fermented by the gut microbiota [55]. In this regard, the results of recent interventional studies suggest that dietary fiber increments may reduce diversity. This is because the microbes that digest fiber become exclusively enriched, resulting in a change in intestinal composition and, through competitive interactions, decreased diversity [56]. Gut bacterial degradation by dietary polysaccharides happens in two phases: (1) internal anaerobic glycolysis and (2) polysaccharides are hydrolyzed extracellularly to produce mono- and disaccharides [57].

Bearing in mind the above mentioned, by increasing the growth of certain intestinal bacteria during intestinal fermentation (among others), polysaccharides can alter the microbiota profile of the intestinal microbiota and change the physiology of the host, both locally and remotely [58].

On the other hand, *Bifidobacterium longum*, an example of bacteria with the ability of microbial fermentation, has the advantage of using the fucosylated oligosaccharides present in human milk to inhibit the growth of specific bacteria such as *Escherichia coli* and *Clostridium perfringens* [59]. In addition, *Bacteroides* species may consume those fucosylated oligosaccharides as a carbon source [60]. Infants born to mothers with nonfunctional fucosyltransferase 2 (FUT2), which is required for the fucosylation of milk oligosaccharides, have low levels of *Bacteroides* and *Bifidobacterium* in their feces [61]. In humans, patients with insulin resistance show elevated levels of *Dorea* and *Coprococcus*. Polysaccharide-containing bacteria possess degradation properties, and their associations with fecal sugar derivatives were generally positive, while *Alistipes* showed a negative correlation [62]. *Dorea* strain administration on a high-fat diet mice intensified insulin resistance and obesity compared with *Alistipes* administration. The authors of this work reported that the gut microbes’ effects on metabolic diseases are mediated through polysaccharides’ microbial fermentation and their derivatives [62]. Several studies in mice involving species of *Bacteroides* have shown that controlling the intake of polysaccharides in the mouse diet allows species selection that are capable of metabolizing the complex glycans present, such as human milk oligosaccharides [60], fructans [63], fucosylated mucin glycans [64] and mannan [65], among others.

For a comprehensive understanding of the effects of polysaccharides on gut health and the host, more detailed information is required. Therefore, the present review aims to elucidate the knowledge of the function played by dietary polysaccharides on gut microbiota composition and how these effects affect host health. We addressed the impact of several polysaccharides in health-promoting effects through the modulation of gut microbiota. Finally, we summarize recently reported studies in the field conducted on humans.

## 2. Health-Promoting Effects of Polysaccharides through the Modulation of Gut Microbiota

### 2.1. Dietary Polysaccharides and Short-Chain Fatty Acids (SCFAs)

Short-chain fatty acids (SCFAs) are metabolites produced by bacteria that can pass through the intestinal barrier and interact with host cells, thereby affecting the immune response [66]. When fiber is anaerobically fermented by gut microbiota, polysaccharides and proteins are metabolized into SCFAs [1]. In Figure 1 we summarize the bacterial degradation of polysaccharides in the intestine by fermentation.

A growing body of evidence suggests that SCFAs are capable of modulating the inflammatory response of immune cells, including neutrophils, dendritic cells, macrophages, monocytes, and T cells [67,68,69].

Obligate anaerobes hydrolyze nondigestible carbohydrates into oligosaccharides, which are fermented in an anaerobic environment. Anaerobes convert hexoses to pyruvate by a process similar to glycolysis before oxidizing pyruvate to acetyl-CoA in conjunction with reduction of an electron carrier or, in many cases, hydrogen gas [70,71]. From there, acetyl CoA is converted into various SCFAs.

As soon as SCFAs are produced, they are absorbed by colonocytes, primarily through sodium-dependent monocarboxylate transporters or H+-dependent monocarboxylate transporters. SCFAs affect intestinal mucosal immunity and influence barrier integrity and function by binding to G protein-coupled receptors, including free fatty acid receptors 2 and 3, as well as GPR109a/HCAR2 and GPR164 [72,73]. In a mouse model of colitis induced by dextran sulfate sodium, SCFAs binding to GPR43 and GPR109A stimulated K+ efflux and hyperpolarization, resulting in NLRP3 inflammasome activation and increased levels of IL-18 in serum [74]. Hence, SCFAs and their receptors contribute to health benefits associated with dietary fiber, as well as the way in which metabolite signals feed through to a major path for gut homeostasis.

SCFAs have an important role in intestinal immune homeostasis maintenance [75,76,77]. By ratifying and purging antigens from neutrophils and monocytes, immunological responses could be triggered, and pathogens could be prevented from invading. [75,76,77].

The normal gut microbiome makes 50–100 mmol·L^−1^ SCFAs per day and works as a source of energy for the host’s gut epithelium [78]. These SCFAs can be rapidly absorbed in the colon and serve many diverse roles in regulating gut inflammation, motility, energy harvesting, and glucose homeostasis [79,80].

The most common SCFAs are acetates, butyrates, or propionates, and a large proportion of these acetates undergo lipogenesis in adipose tissue and undergo oxidization in muscle, while some are converted into butyrates by bacteria [28]. Both butyrate and propionate protect the host from hypertensive cardiovascular damage [81] and butyrates are also associated with intestinal barrier integrity and may have beneficial effects on the epithelium of the gut [82].

### 2.2. Dietary Polysaccharides Influence Immunity by Acting as Prebiotics by Changing Gut Microbiota Composition

Biologically, polysaccharides perform a wide variety of functions and are capable of producing prebiotics that stimulate the microbiota in the intestines. The intestinal microbiota also exerts beneficial effects by selectively degrading polysaccharides, which can be used by the intestinal microbiota as a source of energy to maintain the physiologic effects of the intestinal bacteria and regulate their composition [69]. Some polysaccharides, such as dietary fibers, resist hydrolysis in the stomach and the small intestine of humans. According to Dolan et al., prolonged deficiency of dietary fiber can permanently alter gut microbiota and result in gut dysbiosis [83].

Non-fermentable polysaccharides are excreted in the large intestine while fermentable polysaccharides are digested by the microbiota that inhabits in the large intestine and are fermented to produce diverse metabolites that provide the host with energy [84,85].

Certain polysaccharides act as immunomodulators and influence the regulation of immune responses during the progression of some diseases [86]. Moreover, natural polysaccharides are capable of enhancing immunity by promoting beneficial microorganisms and increasing immune cell function [87].

Sheng et al. have reported that *Hericium erinaceus*-derived polysaccharides can help to restore humoral and cellular immunity in a murine model by improving the phagocytic function of natural killer cells, phagocytes, secretory IgA, and increasing the activity of AKT and MAPK signaling pathways [88]. Several studies have demonstrated that polysaccharides from ginseng can enhance immunity in sows by increasing the levels of interleukin (IL)-2, IL-6, immunoglobulin (Ig)-G, tumor necrosis factor-alpha (TNF-α), and interferon-gamma (IFN-γ) in both milk and serum [89]. Some other polysaccharides isolated from *Robinia pseudoacacia* and young barley leaves have also been shown to enhance IgA-related cytokines, leukocytes, transforming growth factor-beta (TGF-β), and IL-10 levels [87,90,91].

*Bacteroides* possess the ability to degrade dietary polysaccharides, as well as the polysaccharides on the surface of other gut microbes, and this is the major factor that enables them to thrive within the gut environment [77]. These species could metabolize dietary polysaccharides to SCFAs [92].

Polysaccharides isolated from *Artemisia sphaerocephala* might prevent the diversity decrease associated with bacteria belonging to *Proteobacteria* and *Helicobacter* in an animal model of high-fat diet-induced obesity [93]. Also, *Chlorella pyrenoidosa* and *Spirulina platensis* can restructure the gut microbiota in an animal model of obesity using a high-fat diet, increasing beneficial bacteria from *Bacteroidia*, *Clostridia*, and *Mollicutes*, and decreasing some bacteria from *Verrucomicrobia* and *Actinobacteria* [94].

Some reports have shown that alginate in brown seaweed modulates the obesity-related with a high-fat diet by regulating SCFA production and changing the *Bacteroidales* and *Clostridiales* [95]. *Laminaria japonica* soluble polysaccharides diminish non-alcoholic fatty liver diseases in a high-fat diet animal model through decreasing the *Firmicutes/Bacteroidetes* ratio and stimulating *Verrucomicrobia* and propionate-producing bacteria *Akkermansia* (a bacterium of the phylum *Verrucomicrobia*) and *Bacteroides* [94].

Accordingly, *Akkermansia muciniphila* is involved in the metabolism of mucin and the maintenance of intestinal integrity [96]. The increment of *Akkermansia muciniphila* after polysaccharide interventions has been related to benefits to the host (e.g., [97,98,99]). By contrast, other studies define *Verrucomicrobia* phylum as “unfavorable” for the prevention of obesity, and higher levels of this bacteria have been associated with this disease [96]. These discrepancies may be due to the fact that not all subspecies of *Verrucomicrobia* (e.g., *Akkermansia muciniphila*) may display the same specific properties, the model used in the study (animal model, or humans), as well as the basal state of the microbiota (eubiosis or dysbiosis, healthy or not subject, etc.). Overall, the fact is that the bacteria belonging to the phylum *Verrucomicrobia* are widespread contributors to the cycling of carbon and have the capacity for starch degradation is a crucial component of plant biomass [100].

## 3. Human Studies Examining Polysaccharide Modulation of Gut Microbiota and Its Association with Improved Health

The intestinal microbiota plays a vital role in human physiology through the production of metabolites that regulate essential activities that facilitate a symbiotic relationship between the microbes and the host. Polysaccharides are key regulators of colon physiology and the changing intestinal environment [101], and they are selectively used by gut microbiota to enhance the selection, colonization, and survival of probiotic bacteria acting as prebiotics [102].

The consumption of prebiotics is currently increasing, as well as the interest in them as functional foods. Therefore, research aimed at deciphering the mechanisms involved and their precise health effects has augmented exponentially.

In this regard, numerous clinical trials have already been conducted addressing a wide range of diseases (from obesity to chronic kidney disease) through dietary intervention with different polysaccharides. These studies are mainly focused on evaluating the potential of these polysaccharides as modulators of the intestinal microbiota to counteract the detrimental effects of the pathology (Table 1).

A clear example of the latter is inulin, a functional food found naturally in various plants and vegetables, which is a widely used ingredient in diverse efficacy studies thanks to its prebiotic properties [103]. Inulin is being investigated as a potential modulator of the gut microbiota with benefits for human health. The most notable recently reported changes induced by inulin are an increase in *Bifidobacterium,* an improvement in function, as well as benefits in host metabolism for a variety of metabolic diseases, including obesity, type 2 diabetes, kidney disease, intestinal disease, and non-alcoholic fatty liver disease [97,104,105,106,107,108,109,110,111,112,113,114].

In addition to being a dietary fiber beneficial to health, RS is also defined as the portion of starch that cannot be digested or absorbed by humans in their small intestine. By fermenting RS, the gut microbiota can produce SCFAs [115]. In recent years, clinical investigations addressing the use of RS as a microbiome-modifying strategy have proliferated. In this particular case, supplementation with RS in patients with renal disease has led to an elevation in *Faecalibacterium* and a decrease in systemic inflammation [116], as well as elevated SCFA producers’ microbes [107]. Additionally, a SCFA increment after RS intervention has been positively correlated with the relative abundance of *Faecalibacterium*, *Ruminococcus*, *Roseburia*, and *Barnesiellaceae* [117] and is effective in reducing body fat in healthy individuals [98].

The consumption of β-glucans has been shown to reduce calorie intake, lower cholesterol levels, and improve immunity [118]. Moreover, several clinical trials have also shown changes in gut microbiota composition and metabolic parameters. After dietary interventions with this prebiotic, changes in gut microbiota composition related to the increase of healthy bacteria (*Bifidobacterium* and *Akkermansia*) were observed in patients at high risk of developing metabolic syndrome [119,120]. Furthermore, in patients suffering from chronic kidney disease, β-glucan intake significantly altered the levels of the uremic toxin of intestinal origin and improved the state of the intestine [99].

On the other hand, non-invasive therapies such as prebiotic intake are becoming increasingly popular as a means to improve the quality of life of older adults [121]. In this regard, we found studies that examined the impact of polysaccharides on elderly people, but the results were conflicting. For instance, while Kiewiet et al. have reported changes in microbiota that were associated with improvements in health [111], Ganda et al. observed no significant effects following the intervention [122].

## 4. Future Perspectives in the Nutrition Field

Cell plant walls are composed of diverse types of polysaccharides and proteins which play vital roles in biology, including the regulation of cell expansion and tissue attachment, exchange of ions, as well as defense against pathogenic microorganisms. Further, there is evidence that fermentable dietary fiber from polysaccharides has biological activities that are low in toxicity, and have anti-oxidant, anti-inflammatory, anti-tumor, and antiviral effects. Moreover, evidence also suggests that polysaccharides play an active role in the symbiotic relationship between the gut microbiota and the host. Indeed, microbes convert complex polysaccharides into monosaccharides through a variety of biochemical pathways mediated by enzymatic activities. Together with polysaccharides, colonic bacteria also produce lactic acid, which reduces colonic pH and alters gut microbial composition. As immunomodulators, bacterial polysaccharides protect host cells from pathogenic microbial neighbors, and host-derived polysaccharides interact with gut microbes to influence gut health.

However, it is necessary to point out that fibers derived from polysaccharides obtained from different types of plants have different chemical compositions and physicochemical properties. Consequently, plant-based diets will provide a variety of dietary fibers as well as a variety of microbiota compositions. The genetics and the pre-diet microbiome of the host will also add variability to the effects of plant-based diets on microbiota composition.

On the other hand, although many of the physiological and nutritional effects of dietary polysaccharides are widely known, the different mechanisms of action have yet to be fully elucidated, as occurs, for example, with NSP. Along the same line, polysaccharides from marine algae are increasingly being used as prebiotics. These compounds are a rich source of dietary fiber, which are not decomposed by the enzymes of the upper gastrointestinal tract. Polysaccharides from marine seaweeds also have a detoxifying effect. Conversely, other factors, such as the complex chemical structure of some of these products must still be completely understood [138], as well as the high presence of sulfate residues in some of them, which may limit their fermentation by the gut microbiota and increase toxicity.

Several factors may affect the consistency of the results regarding polysaccharides’ effects on the gut microbiota, including methodological sampling and bioinformatic pipelines. To make conclusions regarding this issue, it is necessary to take these factors into account.

To conclude, despite the recent work in this field, we are only beginning to understand how dietary polysaccharides affect health by modulating gut microbiota. Progress is challenged by the wide variety of dietary polysaccharides, their interactions with other molecules such as proteins, and by the vast variations in gut microbiota profiles. In this sense, it is essential to emphasize the importance of carefully selecting a sampling method when analyzing the composition of the microbiota in order to avoid contradictory results and to be able to obtain a solid understanding of all the processes and the precise role of all the “players” involved.

## Figures and Tables

**Figure 1 nutrients-14-04116-f001:**
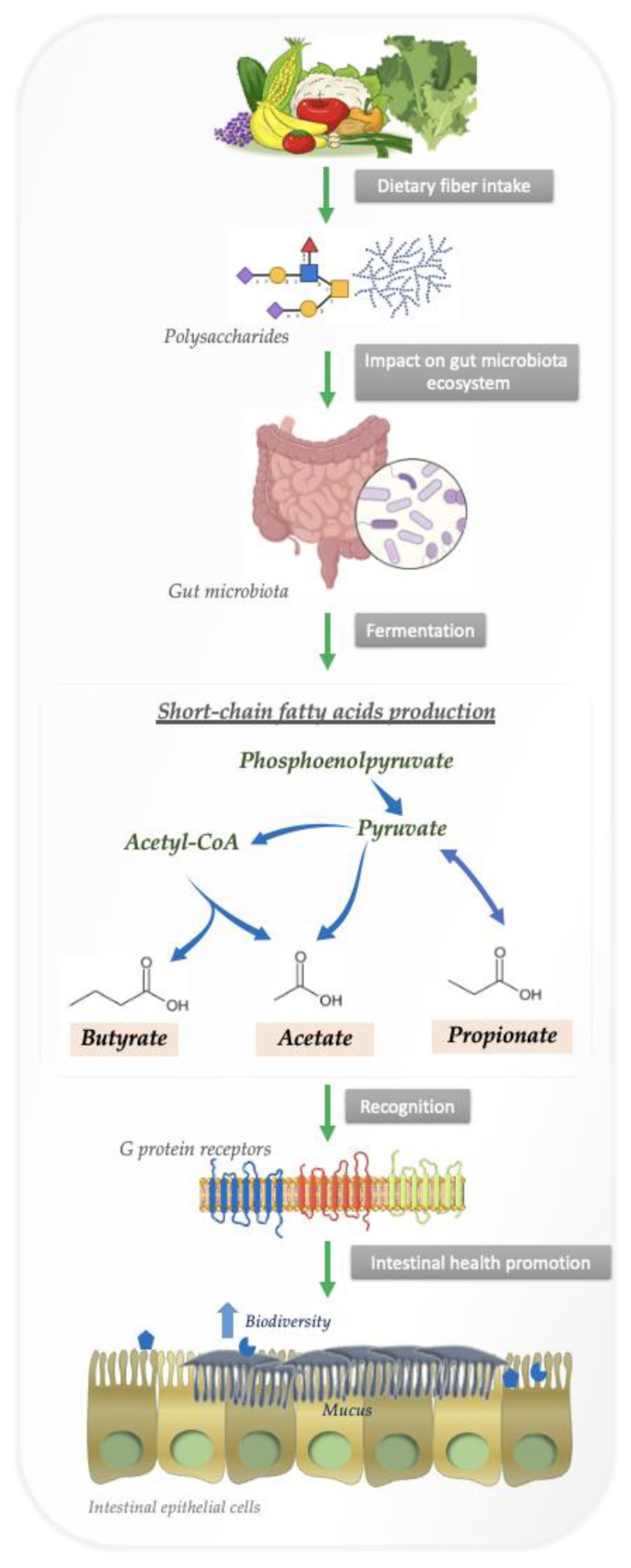
Bacterial degradation of polysaccharides in the intestine by fermentation.

**Table 1 nutrients-14-04116-t001:** Human studies addressing the modulation of gut microbiota by polysaccharides.

Author/Year	Type of Study	Aim	Study Design	Most Remarkable Findings
Nilholm et al., 2022, [123]	RCT	Identify the effects of reduced starch and sucrose consumption on the gut microbiota and circulating microRNA in patients with digestive symptoms	N = 80. IBS patients 4-week SSRD intervention dietary records, and symptom questionnaires; IBS symptom severity score; Visual analog scale for IBS. 16S rRNA sequencing	↑ β diversity associated with changes in nutrient intake, ↓ Gastrointestinal symptoms, ↑ *Eubacterium*, *Lachnospiraceae* UCG-001, *Eggerthella*, ↓*Acidaminococcus*, *Slackia*, *Catenibacterium*
Rodríguez et al., 2022, [104]	RCT	To evaluate the impact of physical activity and prebiotic supplementation in obese subjects	N = 60. Obese subjects, intake of 16 g/day of native inulin plus dietary advice for 3 months and increased physical activity. 16S rRNA sequencing	↓ BMI,↓ liver enzymes and plasma cholesterol, and an improvement in glucose tolerance were observed. *Bifidobacterium*, *Dialister*, and *Catenibacterium* regulations. Improved glucose homeostasis and increased gut fermentation
DeMartino et al., 2022, [124]	RCT	To measure changes in the gut microbiota and fecal SCFAs	N = 50. Healthy adults. BMI 24.5 ± 3.6 kg/m^2^. Daily intake of a side dish containing one potato (averaging 145g) for 4 weeks	Potato dish consumption produced ↓ Alpha diversity ↑ *Hungatella xylanolytica* and *Roseburia faecis*
Ebrahim et al., 2022, [99]	RCT	In this study, a β-glucan prebiotic was examined concerning kidney function, uremic toxins, and gut microbiome	N *=* 3. Chronic kidney disease patients in stages 3 to 5. Intake of 13.5 g/day of β-glucan for 14 weeks. 16S rRNA sequencing	Altered uremic toxin levels of intestinal origin and favorably affected the gut microbiome.
Nolte Fong et al., 2022, [125]	RCT	To predict PPGR. To develop a precision nutrition model to predict PPGR after the intervention of low-versus high-RS-potatoes.	N = 30. Women BMI: 25–40 kg/m^2^. Daily intake of 250 g of hot (9.2 g RS) or cold (13.7 g RS) potatoes. 16S rRNA sequencing	Mostly *Faecalibacterium*, predicted the PPGR, inverse relationships between. Low-RS potato, moderate height and *Faecalibacterium*, inverse relationships between glucose iAUC, insoluble fiber intake, and *Actinobacteria*
Xu et al., 2021, [126]	RCT	The purpose of this study is to examine the relationship between blood lipids, gut microbiota, and plasma SCFAs	N = 210. Chinese population, mild hypercholesterolemia. Intake of 80 g of oats or rice daily for 45 days. Pyrosequencing-based analysis	↓Total cholesterol, ↑ *Akkermansia muciniphila*, *Roseburia*, *Dialister*, *Butyrivibrio*, and *Paraprevotella*, ↓ f-*Sutterellaceae*, Negative correlation between *Bifidobacterium* and low-density lipoprotein cholesterol. Positive correlation between *Enterobacteriaceae*, *Roseburia*, and *Faecalibacterium prausnitzii* and plasma butyric acid
Williams et al., 2022, [105]	RCT	To assess oligofructose-enriched inulin supplementation on the gut microbiome and the peak oxygen uptake response to high-intensity interval training	N = 31. Sedentary and healthy women BMI = 25.9 kg/m^2^, 6 weeks of supervised high-intensity interval training plus 12 g/day of oligofructose-enriched inulin	Greater Improvement in VTs, *Bifidobacterium*, and several metabolic processes related to exercise capacity
Mitchell et al., 2021, [127]	RCT	To determine the efficacy of inulin supplementation in improving glucose metabolism and reducing the risk of type 2 diabetes	N = 24. Adults at risk for T2D, BMI: 31.3 kg/m^2^. Intake of inulin (10 g/day) for 6 weeks. 16S rRNA sequencing	↓ Fasting insulin, ↑*Bifidobacterium*
He et al., 2021, [106]	RCT	To examine the effect of inulin on intestinal microbiota and serum UA levels in end-stage renal disease	N = 62. Continuous ambulatory peritoneal dialysis patients. Intake of inulin-type prebiotics (10 g/day, 12 weeks). Shotgun metagenomics sequencing	↓ Serum UA, ↑ Fecal UA degradation was positively associated with *Firmicutes/Bacteroidetes*, enriched *Clostridium* sp. CAG:7, *C.* sp. FS41, *C. citroniae*, *Anaerostipes caccae*, and *C. botulinum*
Kemp et al., 2021 [107]	RCT	To evaluate the effects of enriched RS-2 cookies on the gut microbiome in hemodialysis patients	N = 20. Hemodialysis patients. Intake of 16 g/day of Hi-Maize 260 for 4 weeks. 16S rRNA sequencing	↓ Pielou’s evenness. ↑ Amplicon Sequencing Variants *Roseburia* and *Ruminococcus gauvreauii*, ↓ *Dialister*
Hedin et al., 2021, [108]	Clinical Trial	To determine if supplementation with oligofructose/inulin impacts the risk phenotype in Crohn’s disease patients and siblings.	N = 19. Patients with inactive Crohn’s disease and 12 of their unaffected siblings. Intake of oligofructose/inulin (15 g/day) for 3 weeks. Fecal microbiota was analyzed by qPCR	↑*Bifidobacterium* and *B. longum* in patients and siblings. ↑ *B. adolescentis* and *Roseburia* spp. only in siblings. ↓ Intestinal permeability in patients similar to siblings. ↓ Blood T cell abundance in siblings but not in patients
Shimada et al., 2021, [128]	RCT	To test if rhamnan sulfate decreases constipation	N = 38. Subjects with low defecation frequencies. Administration of rhamnan sulfate (100 mg/day) for 2 weeks. 16S rRNA sequencing	↑ Frequency of dejection without side effects, functional alternation of the KEGG pathways
Yoon and Michels. 2021, [129]	RCT	To evaluate the effect on the intestinal microbiota composition and function of combined calcium and inulin supplementation, calcium supplementation alone, inulin supplementation alone	N = 12. Healthy adults. Consumption of the three interventions in a random sequence for 4 weeks each intervention, 2 g of calcium powder, 15 g of inulin, or a combination of 2 g of calcium and 15 g (once a day). 16S rRNA sequencing	No differences in microbial composition, short-chain fatty acids concentration, or lipopolysaccharide-binding protein concentrations
Biruete et al., 2021, [130]	RCT	To assess the impact of supplementation of inulin on the gut microbiota composition and microbial metabolites	N = 12. Hemodialysis patients. BMI = 31.6 kg/m^2^. Intake of inulin (10 g/d for females; 15 g/d for males) or maltodextrin [6 g/d for females; 9 g/d for males] for 4 weeks. 16S rRNA sequencing	Inulin ↑ *Verrucomicrobia* and its genus *Akkermansia*, inulin and maltodextrin: ↑ *Bacteroidetes* abundance and its genus *Bacteroides*, ↑ fecal acetate and propionate
Hughes et al., 2021, [117]	RCT	To investigate the effects of RS2 from wheat on glycemic response, its impact on metabolic health, and gut microbiota	N = 30. Healthy subjects, BMI > 18.5 > 39.9 kg/m^2^. Intake of RS2-enriched wheat and wild-type wheat were provided as supplement food for 7 days. 16S rRNA sequencing	↓Postprandial glucose and insulin responses, ↑*Ruminococcus* and *Gemmiger* in the fecal contents, reflecting the composition in the distal intestine. Additionally, fasting breath. Butyrate and total SCFAs were positively correlated with the relative abundance of *Faecalibacterium*, *Ruminococcus*, *Roseburia*, and *Barnesiellaceae*
Morales et al., 2021, [119]	RCT	To evaluate the hypocholesterolemic, immune and microbiota-modulatory effect of a mushroom extract hypercholesterolemic subjects	N = 52. Subjects with untreated mild hypercholesterolemia. Intake of a β-D-glucan-enriched mixture (10.4 g/day) obtained from shiitake mushrooms. 16S rRNA sequencing	No inflammatory or immunomodulatory responses. No changes in IL-1β, IL-6, TNF-α, or oxLDL. A positive association between *Prevotella*_9, *Alistipes* and maltodextrin. In β-D-glucan-enriched mixture the most responsive genera were *Eubacterium ventriosum* group, *Erysipelotrichaceae*_UCG-003, A*kkermansia*, *Coprobacter*, *Lachnoclostridium*, *Bacteroides,* and *Alistipes*
Neyrinck et al., 2021, [109]	RCT	To test if inulin intake influences fecal microbial-derived metabolites and markers related to gut integrity and inflammation in obese patients	N = 24. Obese patients. Intake of 16 g/day native inulin. Dietary advice to consume inulin-rich versus inulin-poor vegetables for 3 months. Caloric restriction. 16S rRNA sequencing	↑ *Bifidobacterium*, ↑ fecal rumenic acid, a conjugated linoleic acid, ↓ calprotectin, both interventions: ↑ the ratio of tauro-conjugated/free bile acids in feces
Leyrolle et al., 2021, [110]	RCT	The purpose of this study is to establish a potential connection between gut microbiota changes and their effects on mood and cognition following inulin intake	N = 106. Obese patients. Intake of 16 g/day of native inulin. Dietary advice to consume inulin-rich or -poor vegetables for 3 months. Caloric restriction. 16S rRNA sequencing	Moderate beneficial effect on emotional competence and cognitive flexibility. Patients exhibiting higher *Coprococcus* levels at baseline were more prone to benefit from prebiotic supplementation. Positive responders toward inulin intervention showed worse metabolic and inflammatory profiles at baseline
Kiewiet et al., 2021, [111]	RCT	To test if chicory long-chain inulin intake changes microbiota composition, microbial fermentation products, and immunity in the elderly	N = 182. Old healthy elderly individuals (55–80 years), Intake of long-chain inulin 8 g/day for 2 months. 16S rRNA sequencing	↑ Microbial diversity, ↑ abundance of the *Bifidobacterium* genus, *Alistipes shahii*, *Anaerostipes hadrus*, and *Parabacteroides distasonis*, ↓ isobutyric acid levels
Berding et al., 2021, [131]	RCT	Efficacy of polydextrose in the improvement of cognitive performance and acute stress responses by manipulation of the gut microbiota in healthy subjects	N = 18. Healthy females. Intake of 12.5 g/day Litesse^®^Ultra (> 90% PDX polymer) for 4 weeks. 16S rRNA sequencing	Polydextrose improved cognitive flexibility. Better performance in sustained attention,↑ abundance of *Ruminiclostridium*, attenuation of adhesion receptor CD62L
Ganda Mall et al., 2020, [122]	RCT	Effect of oat β-glucan and wheat arabinoxylan on the intestinal barrier function and their potential to counteract indomethacin anti-inflammatory induced hyperpermeability in the elderly.	N = 49 elderly subjects (≥65 years). Intake of (12 g/day) of oat β-glucan or arabinoxylan for six weeks. 16S rRNA sequencing	No significant effects were observed after intervention
Reider et al., 2021, [132]	Clinical Trial	To investigate the microbiota-modeling effects of partially hydrolyzed guar gum	N = 20. Healthy subjects. 3 weeks of a lead-in period, three weeks of intervention (5 g partially hydrolyzed guar gum up to three times per day and a three-week washout period. 16S rRNA sequencing	↑ Stool frequency and consistency, ↑ *Ruminococcus*, *Fusicatenibacter*, *Faecalibacterium,* and *Bacteroides*, ↓ *Roseburia*, *Lachnospiracea,* and *Blautia*, ↓ α- diversity
Hiel et al., 2020, [97]	RCT	To evaluate the impact of native inulin on gut microbiota in obese patients	N = 150. Obese patients. Intake of 16 g/day of native inulin. Advice to consume inulin-rich versus -poor vegetables for 3 months, Caloric restriction. 16S rDNA sequencing	↓ Energy intake, BMI, systolic blood pressure, and serum g-GT, ↓ *Desulfovibrio* and *Clostridium sensustricto*, ↑ *Bifidobacterium*
Reimer et al., 2020, [133]	RCT	To examine the effect of two doses of snack bars, comprising chicory root inulin-type fructans, on gut microbiota in healthy adults with habitual low dietary fiber intake	N = 50. Healthy adults with low dietary fiber intake of isocaloric snack bars of either moderate-dose fiber (7 g/day) or control or low-dose fiber (3 g/day). 4 weeks with 4 weeks washout periods. 16S rRNA sequencing	Moderate dose of inulin-type fructans: ↑ *Bifidobacterium*, *Cellulomonas*, *Nesterenkonia,* and *Brevibacterium*, ↓ *Lachnospira,* and *Oscillospira*, Low-dose of inulin-type fructans: ↑ *Bifidobacterium*
Chong et al., 2020, [112]	RCT	To determine if inulin supplementation after brief metronidazole therapy reduces alanine ALT and maintains weight loss after achieving a VLCD in NAFLD patients	N = 62. NAFLD patients following a 4-week VLCD. 12-week, three-arm trial: 400 mg metronidazole twice daily in week 1 then inulin 4 g twice daily or placebo twice daily in week one then inulin or placebo-placebo. 16S rRNA sequencing	After VLCD: ↓ BMI and ALT, ↓ *Firmicutes/Bacteroidetes*, ↓ *Roseburia*, *Streptococcus,* and *Dialister*, ALT further ↓ after metronidazole-inulin treatment
Deeham 2020, [134]	RCT	To test if small differences in the chemical structure of dietary fiber can induce changes in fecal microbiota composition	N = 10. Healthy humans, 4-week dose-escalation intake of RS4. 16S rRNA sequencing	Crystalline and phosphate cross-linked starch structures induced different effects on the microbiome related to the production of propionate or butyrate, ↓ α-diversity
Sasidharan et al., 2019, [135]	Clinical Trial	To evaluate the benefit of prebiotic amylase RS in reducing the incidence of acute radiation proctitis, in patients receiving radiation therapy for cancer of the cervix	N = 104. Patients receiving radical chemo-radiotherapy for cervix carcinoma. Intake of 30 g/day of amylase RS and other digestible starch throughout the course of the external radiotherapy. PCR amplification of some bacterial communities	No significant benefit after the intervention of RS over and above normal diet to patients receiving pelvic radiotherapy
Hess et al., 2020, [113]	RCT	To investigate how calorie restriction combined with dietary fiber affected body weight and gut microbial composition.	N = 116. Overweight or obese subjects, BMI = 28–45 kg/m^2^. Before initiation: energy-restricted weight-loss period, intake of 10 g inulin plus 10 g resistant maltodextrin per day, 500 kcal/day energy-restricted diet, 12 weeks. 16S rRNA sequencing	↓ Systolic and diastolic, blood pressure, ↑ *Parabacteroides* and *Bifidobacterium*, ↑ diversity of gut microbiota
Yasukawa et al., 2019, [136]	RCT	Partially hydrolyzed guar gum affects stools, plasma bile acids, quality of life, and gut microbiota of healthy volunteers with diarrheal tendencies	N = 44. Healthy adults have at least 7 bowel movements per week and at least 50% of their stool falls within the Bristol stool scale values of 5 and 6. Intake of the PHGG 5 g/day for 3 months. 16S rRNA sequencing	Stool form improvement Bristol stool scale was significantly normalized. ↑ *Bifidobacterium*
Hiel et al., 2019, [114]	Clinical trial	Consumption of inulin-type fructan-rich vegetables on gut microbiota, gastrointestinal symptoms, and food-related behavior in healthy individuals	N = 26 healthy individuals, BMI = 20–25 kg/m^2^. A controlled diet based on the intake of Inulin-type fructans (15 g/day) for 2 weeks. 16S rRNA sequencing	↑ *Bifidobacterium*, ↓ unclassified *Clostridiales*, greater satiety, ↓ desire to eat sweet, salty, and fatty food, intestinal discomfort was inversely associated with *Clostridium* cluster IV
Laffin et al., 2019, [116]	RCT	To test if the supplementation with high-amylose maize RS type 2 benefits the gut microbiome and lows systemic inflammation	N = 20. End-stage of chronic kidney disease patients. Intake of 20 g/day of HAM-RS2 for 1 month and first month 25 g/day during the second month. 16S rRNA sequencing	↓ Serum urea, IL-6, TNFα, and malondialdehyde, ↑ *Faecalibacterium*
Zhang et al., 2019, [98]	RCT	To test the effects of RS in normal body weight subjects	N = 19. Subjects with normal body weight. Intake of 40 g high amylose RS2/day, 16S rRNA sequencing	↓ Visceral and subcutaneous fat areas, ↑ acetate and early phase insulin, C-peptide, and glucagon-like peptide-1 secretion, ↓ low-density lipoprotein cholesterol and blood urea nitrogen levels, ↑ *Ruminococcaceae*_UCG-005, ↑ N-acetyl-DL-tryptophan and indole lactic acid
Velikonja et al., 2019, [120]	RCT	Testing whether consumption of barley beta-glucans modifies gut microbiota composition, SCFA production, and metabolic status in patients with metabolic syndrome	N = 43. High risk for metabolic syndrome development or with diagnosed metabolic syndrome subjects. Intake of bread containing 6 g/day of barley beta glucans for 4 weeks. 16S rRNA sequencing	↓ Total plasma cholesterol, ↑ propionic acid, ↓ microbial diversity and richness, ↑ *Bifidobacterium* spp. and *Akkermansia muciniphila* within a cholesterol-responsive group
Sandber et al., 2019, [137]	RCT	Evaluation of the *Prevotella*/*Bacteroides* ratio (PBR) to distinguish between metabolic responders and nonresponders to barley dietary fiber	Healthy subjects splitting based on *Prevotella* and *Bacteroides* before intervention: Highest PBR N = 12; lowest PBR N = 13.; high abundance of both measured bacteria N = 8. BMI < 28 kg/m^2^, 3-day intervention with barley kernel bread (100 g potentially available starch per day), 16S rRNA sequencing	↓ Blood glucose responses to the breakfast independently of *Prevotella/Bacteroides* ratios, highest *Prevotella/Bacteroides* group: ↓ insulin response and IL-6, high abundance of both measured bacteria ↓ hunger sensations

Abbreviations: ALT, alanine aminotransferase; AUC, area under the curves; BMI, body mass index; g-GT, Gamma-glutamyl transferase; IBS, irritable bowel syndrome; KEGG, Kyoto encyclopedia of genes and genomes; NAFLD, non-alcoholic fatty liver disease; PBR, *Prevotella/Bacteroides* ratio; PPGR, postprandial glucose response; SCFAs, short-chain fatty acids; SSRD, sucrose-reduced dietary; rRNA, ribosomal ribonucleic acid; RS, resistant starch; RCT, randomized controlled trial; T2D, Type 2 diabetes; UA, uric acid; VLCD, very-low-calorie diet.

## Data Availability

Not applicable.

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
