# Peer review of "Dietary Polysaccharides as Modulators of the Gut Microbiota Ecosystem: An Update on Their Impact on Health"

_nutrients, 2022, doi:10.3390/nu14194116_

Round 1
Reviewer 1 Report (Previous Reviewer 1)
This article summarizes the regulatory effects of dietary polysaccharides on the gut microbiota and the health effects on the host. However, this review is too broad. Several questions are listed as following:
1. It is recommended to delete the keyword “human studies”. The scope of this keyword is too general and wide.
2. Line 39-40, The authors listed some polysaccharides commonly found in dietary products, but “starch” appears twice here.
3. Line 41-48, this paragraph focuses on the differences between different types of polysaccharides. The viscosity and fermentability properties of polysaccharides are worth discussing because these physical properties are important for the physiological function.
4. It is recommended to supplement the source of polysaccharides in the part of “1.1. Food polysaccharides, an overview”.
5. Redrawing Figure 1, Figure 1 provides the reader with very little useful information, and the picture of “Dietary polysaccharides intake”, what does the picture on the left mean?
6. It is recommended to add a mechanism figure of short-chain fatty acids in the host.
7. Line 199-216, the content of this paragraph is not related to “immunity”. However this paragraph is entitled "Dietary polysaccharides affect immunity by changing gut microbiota composition".
8. The most obvious problem with this review is that the content is too broad and the topic of discussion is not focused enough. It is recommended that the authors focus on only one health benefit of polysaccharides, such as immune regulation.
Author Response
Ms. Fifteen Hu
Section Managing Editor,
Thank you for providing us with the opportunity to submit a revised version of our manuscript entitled “Dietary Polysaccharides as Modulators of the Gut Microbiota Ecosystem. An Update on their Impact on Health” to the Nutrients journal in the Special Issue titled “Dietary Polysaccharides and the Gut Microbiota Ecosystem”. We would like to thank the reviewers for their thoughtful comments and suggestions regarding our manuscript. All comments have been considered and incorporated into the revised manuscript. Changes are highlighted in green and blue font for the reviewer's comments with an itemized point-by-point response to the reviewers' comments.
COMMENTS FROM REVIEWER #1
The authors would like to thank the reviewer for his/her respected comments and effort made during the review process, which are highly appreciated.
Point-to-point response:
This article summarizes the regulatory effects of dietary polysaccharides on the gut microbiota and the health effects on the host. However, this review is too broad. Several questions are listed as following:
- It is recommended to delete the keyword “human studies”. The scope of this keyword is too general and wide.
Response: Thanks to the reviewer for his/her kind comment, we have deleted this keyword.
- Line 39-40, The authors listed some polysaccharides commonly found in dietary products, but “starch” appears twice here.
Response: Thanks to the reviewer for his/her comment about the word duplication, we have deleted the repeated word.
- Line 41-48, this paragraph focuses on the differences between different types of polysaccharides. The viscosity and fermentability properties of polysaccharides are worth discussing because these physical properties are important for physiological function.
Response: Thanks to the reviewer for his/her kind comment about that paragraph, we have added more detailed information and the manuscript now states (page 2 lines 55-67):
“The ability of dissolved polysaccharides to thicken solutions and form gels is one of their most critical functional characteristics both in terms of formulation functionality and health-related functionality [10]. A large hydrodynamic volume of polysaccharides results in increased viscosity at low concentrations, while a small hydrodynamic volume results in decreased viscosity. As a consequence, high solubility (i.e., favorable interaction with the solvent which results in the polysaccharide expanding and a higher hydrodynamic volume) is beneficial for thickening [11]. Additionally, associative interactions may enhance the thickening properties of some modified starches and celluloses. A polysaccharide's ability to form gels is dependent on its solubility. This is essential to the gel structure's ability to hold water and the formation of a continuous network in the solution. It is therefore necessary for the polysaccharides to interact in some way in order to form associations [12]. In order for molecules to associate, they must exhibit either a hydrophobic effect, partial local crystallization, calcium bridges, or double or triple helices [2].”.
- It is recommended to supplement the source of polysaccharides in the part of “1.1. Food polysaccharides, an overview”.
Response: Thanks to the reviewer for his/her kind comment about that paragraph, we add more detailed information and the manuscript now states (page 1 lines 36-42), “Food products contain polysaccharides derived from many sources, including farms, forests, oceans, fermentation vats, and chemical modification of natural polysaccharides, such as cellulose and starch [5]. From the source and type of source, examples include algal (seaweed extracts) derived from agar, algins, carrageenans, and furcellaran, higher in-soluble plants derived from cellulose, fruit extracts derived from pectin, corn starches, rice starches, wheat starches, beta-glucans, guar gum, locust bean gum, tara gum, psyllium seed gum, and tamarind seed polysaccharides [6].”
- Redrawing Figure 1, Figure 1 provides the reader with very little useful information, and the picture of “Dietary polysaccharides intake”, what does the picture on the left mean?
Response: The figure has been redrawn. According to the previous version of figure 1, the reviewer finds the intake of dietary polysaccharides, first to the left, the chemical structure of polysaccharides (the main skeleton, monomer), and then to the right, the structure as a polymer.
- It is recommended to add a mechanism figure of short-chain fatty acids in the host.
Response: The mechanism short-chain fatty acids mechanism in the host has been added.
- Line 199-216, the content of this paragraph is not related to “immunity”. However, this paragraph is entitled "Dietary polysaccharides affect immunity by changing gut microbiota composition".
Response: Based on the reviewer's comments, this paragraph has been moved to another section related to gut microbiota rather than immunity in order to fit properly.
- The most obvious problem with this review is that the content is too broad and the topic of discussion is not focused enough. It is recommended that the authors focus on only one health benefit of polysaccharides, such as immune regulation.
Response: We regret the reviewer's comments about our manuscript’s topic. In section 3, we discussed the role played by dietary polysaccharides on gut microbiota composition and how these effects affect host health. This issue has been highlighted by human studies performed in recent years (Table 1, most remarkable findings). Maybe an additional search could be conducted in preparation for a future manuscript.

Reviewer 2 Report (Previous Reviewer 2)
Authors have submitted a very thorough and well written review on benefits of dietary polysaccharides on the gut microbiota. They addressed my previous concerns adequately.
Author Response
Ms. Fifteen Hu
Section Managing Editor,
Thank you for providing us with the opportunity to submit a revised version of our manuscript entitled “Dietary Polysaccharides as Modulators of the Gut Microbiota Ecosystem. An Update on their Impact on Health” to the Nutrients journal in the Special Issue titled “Dietary Polysaccharides and the Gut Microbiota Ecosystem”. We would like to thank the reviewers for their thoughtful comments and suggestions regarding our manuscript. All comments have been considered and incorporated into the revised manuscript. Changes are highlighted in green and blue font for the reviewer's comments with an itemized point-by-point response to the reviewers' comments.
COMMENTS FROM REVIEWER #2
The authors would like to thank the reviewer for his/her respected comments and effort made during the review process, which are highly appreciated.
Point-to-point response:
Authors have submitted a very thorough and well written review on benefits of dietary polysaccharides on the gut microbiota. They addressed my previous concerns adequately.
Response: We would like to thank the reviewer for his/her kind comments regarding our previous work on the concerns.

This manuscript is a resubmission of an earlier submission. The following is a list of the peer review reports and author responses from that submission.
Round 1
Reviewer 1 Report
This article summarizes the regulatory effects of dietary fiber polysaccharides on the gut microbiota and the health effects on the host. Several questions are listed as following:
1. In the introduction, what is the purpose of mentioning phenolic acids from cereal grains?
2. The layering of the whole article is not clear. It is recommended to add secondary headings to accurately divide the content.
3. In Figure 1, the picture of “Dietary polysaccharides intake”, what does the picture on the left mean?
4. It is recommended to supplement the source and structural characteristics of polysaccharides with healthy effects.
Author Response
Ms. Fifteen Hu
Section Managing Editor,
Thank you for providing us with the opportunity to submit a revised version of our manuscript entitled “Health and the Gut Microbiota Ecosystem as a result of Dietary Polysaccharides” to the Nutrients journal in the Special Issue titled “Dietary Polysaccharides and Gut Microbiota Ecosystem”. We would like to thank the reviewers for their thoughtful comments and suggestions regarding our manuscript. All comments have been considered and incorporated into the revised manuscript. Changes to the original document are tracked for repetition rate and highlighted in green and blue font for the reviewer's comments, with an itemized point-by-point response to the reviewers' comments.
COMMENTS FROM REVIEWER #1
The authors would like to thank the reviewer for his/her respected comments and effort made during the review process, which are highly appreciated.
Point-to-point response:
1.- This article summarizes the regulatory effects of dietary fiber polysaccharides on the gut microbiota and the health effects on the host. Several questions are listed as following:
In the introduction, what is the purpose of mentioning phenolic acids from cereal grains?
Response: The purpose of this study is to demonstrate that dietary polysaccharides (cereal grain) can be used in food technology as bioactive compounds, and phenolic acid is an example of such a compound.
The text has been updated with additional information, and now states (page 2, lines 53-61): “The majority of phenolic acids found in cereal grains are bound to cell walls, with the free form concentration being approximately 100μg/g. However, this concentration in-creases in germinated and malted grains. In the large intestine, enzymes derived from microbes (e.g., esterases and hydroxylases) release 70–95% of the phenolic compounds (which are covalently linked to cell wall NSPs) [9]. It should be noted that cereal bran is a major source of phenolic acids (antioxidants), fibers, and minerals [10]. Aleurone, how-ever, is the critical component that is generally overlooked in favor of indigestible fibers. Additionally, it contains the highest amount of bioactive substances, with ferulic acid being the most significant antioxidant [11,12]”
2.- The layering of the whole article is not clear. It is recommended to add secondary headings to accurately divide the content.
Response: We have added the secondary headings based on the reviewer's comments.
3.- In Figure 1, the picture of “Dietary polysaccharides intake”, what does the picture on the left mean?
Response: According to figure 1, the reviewer finds the intake of dietary polysaccharides, first to the left, the chemical structure of polysaccharides (the main skeleton, monomer), and then to the right, the structure as a polymer.
4.- It is recommended to supplement the source and structural characteristics of polysaccharides with healthy effects.
Response: According to the reviewer's comments, we have amended that paragraph to include new information about the health benefits and now states (page 4, lines 185-202): “Biologically, polysaccharides perform a variety of functions and are capable of producing prebiotics that stimulate the microbiota in the intestines. As well, the intestinal microbiota exerts beneficial effects by selectively degrading polysaccharides, which can be used by the intestinal microbiota as a source of energy to maintain the physiologic effects of the intestinal bacteria and regulate their composition [50].
Glycemic index foods include refined-grain products, white bread, and potatoes, whereas low glycemic index foods include whole-grain products, legumes, and fruits. Postprandial blood glucose and insulin concentrations are greatly influenced by the rate and extent of starch digestion in the intestinal lumen [51]. For individuals with abnormalities in blood glucose regulation, particularly those with type 2 diabetes or metabolic syndrome, the glycemic response to a food is especially important [8,52].
Besides being sources of soluble fiber, polysaccharides bind to bile acids in the small intestine, thereby lowering serum cholesterol and normalizing blood lipid levels [53].
Lipopolysaccharide (LPS) and SCFA are metabolites produced by bacteria that can pass through the intestinal barrier and interact with host cells, thereby affecting the immune response [54]. A growing body of evidence suggests that SCFAs are capable of modulating the inflammatory response of immune cells, including neutrophils, dendritic cells (DCs), macrophages, monocytes, and T cells [50,55,56]”.

Reviewer 2 Report
The submitted narrative review is a well written summary of newer investigations pertaining to the benefits of dietary polysaccharides on the gut microbiota. Strengths of this review include a nicely structured table that summarizes recent human studies and the readability of the text. The introduction provides a nice summary describing various dietary polysaccharides. The breadth of this topic is expansive- therefore, the need for a shallow description of some of the data is understandable. That said, the impact of the manuscript might benefit from the expansion of the discussion in a few areas as described below:
1) For clarification, (Line 249) - Verrucomicrobia is described as harmful, but several studies included in line 255 and in the table (references 107, 114 and 118) cite increases in Akkermansia with polysaccharide interventions as a benefit for the host. The readers might benefit from a discussion as to why we see different health outcomes with this organism. As Akkermansia mainly utilizes host mucin as a carbon source, the certain studies describe the polyphenol component of plants as important to growth of this organism.
2) As a slight continuation of the previous point, the phenolic components of plant materials are highlighted in both the introduction and in the discussion but contribution to host health isn’t specifically addressed in the discussion of the literature.
3) Studies addressing TMAO and metabolites of phosphatidylcholine seem outside the theme of the paper.
4) Perhaps I missed this, but you might consider discussing metabolism more in detail- describing primary degradation and pyruvate metabolism as the benefits of SCFAs are highlighted in your manuscript.
4) Defining specific dietary terms will allow manuscript to be more accessible to a wider audience:
a. (Line 65) Micronutrient density
b. (Line 68) Atwater factors
5) (Line 351) ‘complex chemical structure and the high presence of sulfate residues’ please provide a reference
Very Minor non-content suggestion: For the introduction, there are many small paragraphs, some just consisting of one sentence. Consider consolidation of sentences within larger paragraphs.
Author Response
Ms. Fifteen Hu
Section Managing Editor,
Thank you for providing us with the opportunity to submit a revised version of our manuscript entitled “Health and the Gut Microbiota Ecosystem as a result of Dietary Polysaccharides” to the Nutrients journal in the Special Issue titled “Dietary Polysaccharides and Gut Microbiota Ecosystem”. We would like to thank the reviewers for their thoughtful comments and suggestions regarding our manuscript. All comments have been considered and incorporated into the revised manuscript. Changes to the original document are tracked for repetition rate and highlighted in green and blue font for the reviewer's comments, with an itemized point-by-point response to the reviewers' comments.
COMMENTS FROM REVIEWER #2
The authors would like to thank the reviewer for his/her respected comments and effort made during the review process, which are highly appreciated.
Point-to-point response:
The submitted narrative review is a well-written summary of newer investigations pertaining to the benefits of dietary polysaccharides on the gut microbiota. Strengths of this review include a nicely structured table that summarizes recent human studies and the readability of the text. The introduction provides a nice summary describing various dietary polysaccharides. The breadth of this topic is expansive- therefore, the need for a shallow description of some of the data is understandable. That said, the impact of the manuscript might benefit from the expansion of the discussion in a few areas as described below:
- For clarification, (Line 249) - Verrucomicrobia is described as harmful, but several studies included in line 255 and in the table (references 107, 114 and 118) cite increases in Akkermansia with polysaccharide interventions as a benefit for the host. The readers might benefit from a discussion as to why we see different health outcomes with this organism. As Akkermansia mainly utilizes host mucin as a carbon source, the certain studies describe the polyphenol component of plants as important to growth of this organism.
Response: Thanks for the appropriate observation. To avoid misunderstanding we have deleted the adjective harmful to describe Verrucomicrobia and included a brief paragraph discussing why we see different health outcomes as follows (page 7, lines 294-308): “Accordingly, Akkermansia muciniphila, is involved in the metabolism of mucin and the maintenance of intestinal integrity [105]. The increment of Akkermansia muciniphila after polysaccharide interventions has been related to the benefit of the host (e.g. [106-108]). By contrast, other studies define Verrucomicrobia phylum as “unfavorable” for the prevention of obesity and higher levels of this bacteria have been associated with this disease [105]. These discrepancies may be due to the fact that not all subspecies of Verrucomicrobia (e.g. Akkermansia muciniphila) may display the same specific properties, the model used in the study (in vitro, animal model, or humans) as well as the basal state of the microbiota (eubiosis or dysbiosis healthy or not subject, etc). Overall, the fact is that the bacterial belonging to the phylum Verrucomicrobia are widespread contributors to the cycling of carbon have the capacity for starch degradation a crucial component of plant biomass [109]. On the other hand, Akkermansia muciniphila responds positively to polyphenols and maybe, at least partially, be responsible for the observed health benefits [110]. In fact, the use of polyphenols as a substrate for beneficial bacteria could represent a good strategy to grow them”.
- As a slight continuation of the previous point, the phenolic components of plant materials are highlighted in both the introduction and in the discussion but the contribution to host health isn’t specifically addressed in the discussion of the literature.
Response: The purpose of this study is to demonstrate that dietary polysaccharides (cereal grain) can be used in food technology as bioactive compounds, and phenolic acid is an example of such a compound. The text has been updated with additional information, and now states (page 2, lines 53-61): “The majority of phenolic acids found in cereal grains are bound to cell walls, with the free form concentration being approximately 100μg/g. However, this concentration in-creases in germinated and malted grains. In the large intestine, enzymes derived from microbes (e.g., esterases and hydroxylases) release 70–95% of the phenolic compounds (which are covalently linked to cell wall NSPs) [9]. It should be noted that cereal bran is a major source of phenolic acids (antioxidants), fibers, and minerals [10]. Aleurone, how-ever, is the critical component that is generally overlooked in favor of indigestible fibers. Additionally, it contains the highest amount of bioactive substances, with ferulic acid being the most significant antioxidant [11,12]”
- Studies addressing TMAO and metabolites of phosphatidylcholine seem outside the theme of the paper.
Response: Thanks for the observation. This article summarizes the regulatory effects of dietary polysaccharides on the gut microbiota and the health effects on the host. We remark on human studies addressing the modulation of gut microbiota by polysaccharides. Given that TMAO and metabolites of phosphatidylcholine are an expansive topic per se we have briefly mentioned this point in the present manuscript, and now states, (pages 6, line 235): “Several recent reviews have faced this topic more in detail [76-81].”
4) Perhaps I missed this, but you might consider discussing metabolism more in detail- describing primary degradation and pyruvate metabolism as the benefits of SCFAs are highlighted in your manuscript.
Response: Thanks for the suggestion. We have changed the paragraph to include the information and now it states as follows (pages 3-4, lines 150-156): “As a result of neoglucogenesis, glucose is synthesized from non-carbohydrate carbon substrates such as pyruvate, lactate, glycerol, amino acids derived from glucogenesis, and fatty acids [39].
The Wood-Ljungdahl pathway and acetyl-CoA can be used to produce acetate [40]. There are two ways in which lactobacilli can produce SCFAs: (i) through fermentation of carbohydrates end-products, such as pyruvate, which is produced during the glycolytic pathway; and (ii) by phosphoketolase, under heterofermentation conditions [16,39].”
- Defining specific dietary terms will allow the manuscript to be more accessible to a wider audience:
(Line 65) Micronutrient density
Response: Thanks for the suggestion. We have changed the paragraph to include the definition and now it states as follows (page 2, lines 69-73): “The consumption of sugar can influence micronutrient density (referred to as the concentration of micronutrients mainly vitamins and minerals as well as phytochemicals, essential fatty acids and essential amino acids per calorie of food) [15]. In such situations, complex carbohydrates, particularly those from whole grains, should be increased [1].”
Reference, Adam Drewnowski PhD (2009) Defining Nutrient Density: Development and Validation of the Nutrient Rich Foods Index, Journal of the American College of Nutrition, 28:4, 421S-426S, DOI: 10.1080/07315724.2009.10718106.
(Line 68) Atwater factors
Response: Thanks for the suggestion. We have changed the paragraph to include the definition and now it states as follows (page 2, lines 74-75): “According to Atwater general factor system, a system for assigning energy values to foods [16]”
Reference: The Oxford Dictionary of Sports Science & Medicine (3 ed.)
Publisher: Oxford University Press
Print Publication Date: 2006
Print ISBN-13:9780198568506
DOI: 10.1093/acref/9780198568506.001.0001
eISBN:9780191727788
5) (Line 351) ‘complex chemical structure and the high presence of sulfate residues’ please provide a reference.
Response: The required reference (number 148) has been added (line 403).
Reference: Gotteland, M., Riveros, K., Gasaly, N., Carcamo, C., Magne, F., Liabeuf, G., Beattie, A., & Rosenfeld, S. (2020). The Pros and Cons of Using Algal Polysaccharides as Prebiotics. Frontiers in nutrition, 7, 163. https://doi.org/10.3389/fnut.2020.00163.
6) Very Minor non-content suggestion: For the introduction, there are many small paragraphs, some just consisting of one sentence. Consider consolidation of sentences within larger paragraphs.
Response: Thank you for the advice. when it was appropriate, so as not to mix concepts or ideas, the short paragraphs were consolidated into large paragraphs.
